# The Species Composition and Distribution Patterns of Non-Native Fishes in the Main Rivers of South China

Dang En Gu [1,2,3], Fan Dong Yu [1,2,3], Yin Chang Hu [2], Jian Wei Wang [1,*], Meng Xu [2], Xi Dong Mu [2], Ye Xin Yang [2], Du Luo [2], Hui Wei [2], Zhi Xin Shen [4], Gao Jun Li [4], Yan Nan Tong [4] and Wen Xuan Cao [1]

[1] Institute of Hydrobiology, Chinese Academy of Sciences, Wuhan 430072, China; gudangen@163.com (D.E.G.); yufandong2020@ihb.ac.cn (F.D.Y.); wxcao@ihb.ac.cn (W.X.C.)
[2] Pearl River Fisheries Research Institute, Chinese Academy of Fishery Sciences, Key Laboratory of Recreational Fisheries, Ministry of Agriculture and Rural Affairs, Guangzhou 510380, China; huyc22@163.com (Y.C.H.); xm0557@126.com (M.X.); muxd1019@163.com (X.D.M.); yangyexin@163.com (Y.X.Y.); luodu2012@163.com (D.L.); weihui0630@163.com (H.W.)
[3] University of Chinese Academy of Sciences, Beijing 100049, China
[4] Hainan Academy of Ocean and Fisheries Sciences, Haikou 570100, China; shen_266@msn.com (Z.X.S.); 150599790@163.com (G.J.L.); tongyn@hnhky.cn (Y.N.T.)
[*] Correspondence: wangjw@ihb.ac.cn; Tel.: +86-27-6878-0033

**Abstract:** Non-native fish invasions are among the greatest threats to the sustainability of freshwater ecosystems worldwide. Tilapia and catfish are regularly cultured in South China which is similar to their climate in native areas and may also support their invasive potential. We systematically collected fish from eight main rivers of South China, from 2016 to 2018, to investigate and analyse species' composition and the distribution of non-native fishes. The data reveal that non-native fishes are widespread and abundant in the sampled rivers: of the 98,887 fish collected, 11,832 individuals representing 20 species were not native, which were distributed in the 96% sampled sites. Of the non-native fish species, 17 are used in aquaculture and 19 are native to the tropics; 13 are omnivores while the other seven are predators. Based on dissimilarity of the non-native fish species distributions across the eight rivers, the different rivers could be divided into four assemblages. Geographical isolation and temperature were identified as affecting the distribution patterns of non-native fishes, thereby influencing fish species composition, species number, dominant species, and distribution variations in the South China rivers. Species composition of the non-native fishes in these rivers are related to their introduction vector, compatibility with their native habitat, and feeding strategies. Their distribution was mainly influenced by geographical location and temperature. To mitigate the impacts of non-native fish, a series of stricter management practices, systematic monitoring, and more research are needed.

**Keywords:** biological invasions; freshwater ecosystems; species composition; spatial pattern

---

## 1. Introduction

Non-native species are a primary threat to global biodiversity and the economy [1]. They can cause biodiversity loss, food web impacts, economic damage through competitive interactions, predation, parasite transmission, and habitat alteration [2–5]. Freshwater ecosystems are the most heavily impacted by non-native species, and the most frequent alien group is fish [6–9]. Fishers, fisheries scientists, and natural resource managers must face the complex task of minimizing any economic and ecological losses due to alien fish invasions [10–13].

Thousands of freshwater fish species around the globe have been introduced to new environments, either intentionally or inadvertently [14]. In China, hundreds of non-native fish species are used for aquaculture and the ornamental trade [9,15]. Moreover, rapid trade and economic development in China has placed increasing attention on the need to prevent the establishment of invasive species [16,17]. To date, at least 439 non-native freshwater fish species, representing 22 orders, 67 families, and 256 genera, have been recorded in Chinese freshwaters [8]. A portion of the alien fish species have established self-sustaining populations in nature, though fewer have become invasive; even so, successful fish invaders pose serious threats to the country's economy, human health, and native species [9,18,19].

The distribution of non-native species in China has a clear geographic and ecological bias, as they appear to be prevalent in the southern provinces, particularly Guangdong, Hainan, Guangxi, and Fujian [12,19]. South China accounts for about 81% of the country's total aquaculture production of non-native fish, and is a significant trading center for ornamental fish; numerous non-native fish species have been introduced, farmed, sold, discarded, or have escaped in the region [8,9]. The relatively warm climate and complex network of rivers that allow aquaculture production of non-native fishes also supports their survival and spread in natural waters [13,20]. According to field records and past distribution data on non-native aquatic species, provided by China's Ministry of Agriculture and Rural Affairs, approximately 40 non-native fish species are commonly found in natural freshwaters across China [15,18,19,21]. However, more than 10 non-native fish species are frequently found in the rivers of South China and many of them can be considered successful invaders, including the Nile tilapia *Oreochromis niloticus* (Linnaeus), redbelly tilapia *Coptodon zillii* (Gervais), North African catfish *Clarias gariepinus* (Burchell), mrigal carp *Cirrhinus mrigala* (Hamilton), jaguar cichlid *Parachromis managuensis* (Günther), and suckermouth catfish *Hypostomus* sp. [9,22–25]. To date, systematic monitoring and analysis of the species composition of non-native fishes in South China are still lacking, including determination of the species composition and distribution patterns in particular rivers [9,19].

Invasion by a non-native species progresses in four stages: introduction, establishment, spread, and impact [26]. This process is influenced by the biological characteristics of a species and the environmental conditions in the new habitat [1,27]. For effective establishment, the non-native fish must survive and be dispersed by either natural or anthropogenic means [12,28]. The introduction and spread of a non-native fish species usually relates to human activity, and different vectors involve species with different characteristics, the frequency of introductions, number of discrete release events, and number of released individuals also differs, which can be illustrated as "propagule pressure" [7–9,17,29]. Furthermore, non-native species must overcome multiple barriers to dispersal and ultimately survive in a novel environment, especially as regards the food resources and abiotic conditions of the new habitat (e.g., temperature, salinity, dissolved oxygen, hydrological regime) [1,9,26,30]. Although some successful fish invaders can present a rapid realized niche shift in the invaded range, many non-native species survive most easily in an ecosystem that is similar to their native habitat [31,32]. Furthermore, a non-native species' trophic position influences their survival capacity outside their native range in a context of limited food resources [33–35].

Fish distribution patterns are depended on variations of species composition and biomass, which are typically impacted by their biological characteristics, the existing environmental conditions, and regional socioeconomic factors [11,36,37]. While biotic interactions influence fish abundance and assemblage structure in streams, they seem to play only a small role in comparison to the abiotic conditions [38]. For many non-native fish species, geographical condition and temperature not only determine their introduction, culture, and trade, but also determine their survival, predation, growth, reproduction, and spread in natural waters [21,25,29,30,39,40].

In the present study, we recorded the species composition and distribution patterns of non-native fish species, and explored the potential factors underlying variation in their composition, in eight main rivers of South China, using survey data collected from 2016 to 2018. Specifically, we aimed to:

1) determine what characteristics have promoted the dispersal and survival of non-native fish species, including their introduction vector, native habitat, and feeding strategies; 2) determine differences in the distribution patterns of non-native fish species across the rivers; and 3) highlight whether geographic isolation and temperature strongly supports the non-native fishes' distribution and biomass variation besides the biotic environmental and socioeconomic factors.

## 2. Materials and Methods

### 2.1. Sampling Locations

Fish were surveyed from 2016 to 2018 at 50 sites on rivers in three provinces (Hainan, Guangdong, and Guangxi) of South China. Fish were systematically collected from eight main rivers: Wanquanhe River (WQH), Nandujiang River (NDJ), Changhuajiang River (CHJ), Jianjiang River (JJ), Moyangjiang River (MYJ), Dongjiang River (DJ), Beijiang River (BJ), and Xijiang River (XJ) (Figure 1). The river lengths vary from 160 to 2100 km and the catchments from 3693 to 353120 km$^2$ (Table 1) [41]. The number of sampling sites in each river ranged from 3 to 10 according the previous distribution data on non-native aquatic species, provided by China's Ministry of Agriculture and Rural Affairs (Table 1). Each of the basins of the eight rivers has a warm climate, abundant rainfall, diverse native fish species, and flourishing aquaculture operations [42]. However, the WQH, NDJ and CHJ are located in the tropics and have relatively warmer temperatures [42]. Furthermore, the study rivers are considered to suffer severe invasions of non-native fish species [19,21]. The WQH, NDJ, and CHJ located in the Hainan island belonging to the "Hainan rivers," the JJ and MYJ, which are located in the coastal area of the continent belongs to the "Continent coastal rivers" the XJ, DJ, and BJ belong to the "Pearl rivers."

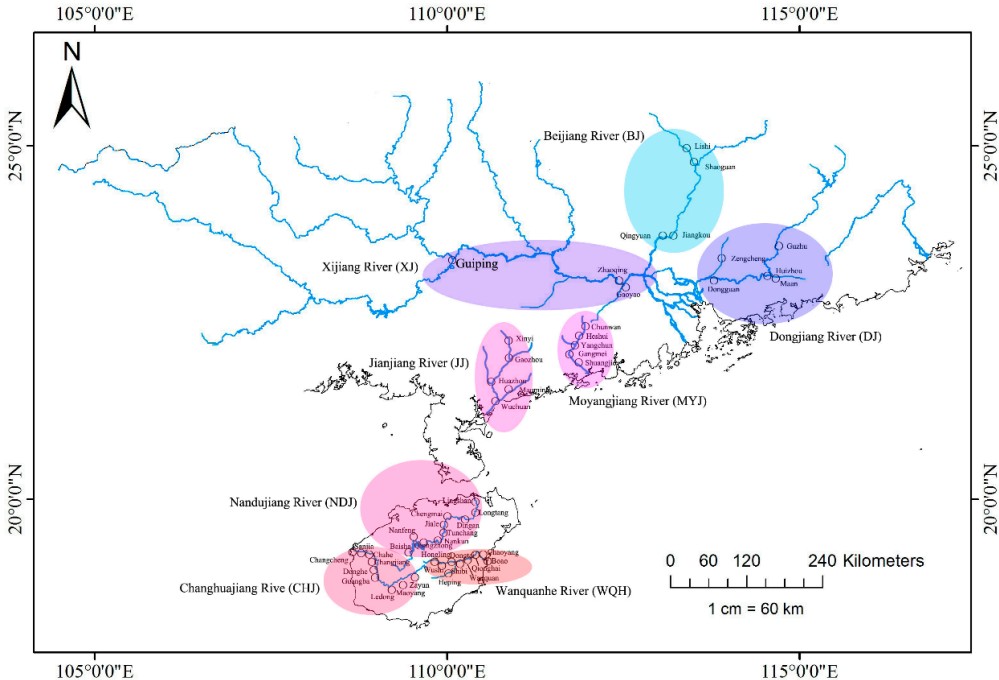

**Figure 1.** Map showing the eight rivers and sampling locations in South China surveyed for fish species composition.

### 2.2. Fish Collection and Data Collection

Field surveys at each site were carried out two times between May to August every year along the latitude, once in May and June, once in July and August. Sampling at each site continued until no new species were caught, which was at least three days. Most of the fish were collected by commercial fisherman, and the remainder by the authors. Field sampling practices followed the "Monitoring

manual of fish resources in the Yangtze River" [43] and the "Standards and specifications on the monitoring of alien fish species" (Ministry of Agriculture and Rural Affairs, China). Fish were primarily caught using gillnets (3 cm mesh) and shrimp nets, and then identified to species, counted and weighted using an electronic scale ($\pm$ 0.01g), following the methods of [19,29,41].

The following data were recorded for each river:

(1)   $W_T$, total weight of all fish collected (kg);
(2)   $N_T$, total number of all fish collected;
(3)   $W_{NT}$, total weight of all non-native fish collected (kg);
(4)   $N_{NT}$, total number of all non-native fish collected;
(5)   $W_{1-I}$, total weight of each fish species (kg), where 'I' represents the number of fish species represented;
(6)   $N_{1-I}$, total number of each fish species collected, where 'I' represents the number of fish species represented;
(7)   $WN_{1-X}$, total weight of each non-native fish species collected (kg), where 'X' represents the number of non-native fish species;
(8)   $NN_{1-X}$, total number of non-native fish species collected, where 'X' represents the number of non-native fish species;
(9)   $FO_{1-I}$, frequency of occurrence of the different fish species, according to the following equation $FO_I = 100 \times (S_I/S)$, where "$S_I$" is the sites number where this species occurs, "S" is the sites number surveyed, 'I' represents the number of fish species represented;
(10)  T, the mean minimum temperature of the eight rivers. These data were obtained from http://lishi.tianqi.com; the mean temperature over 31 days in January 2018 was recognized as the mean minimum temperature of the coldest month for the analysis for each site. After that, the mean temperature of the main sites across each river was recognized as the mean minimum temperature for each of the eight rivers.
(11)  The feeding strategy, introduction vectors, and the native regions of the non-native fish species were collected from the specialized database (National Fisheries Science Data Center of China) and the publications [15,18,19]

*2.3. Data Analysis*

The index of relative importance (IRI) was used to represent the dominance of different fish species. This index is a composite measure that reduces bias in descriptions of animal dietary data but has been proposed as a standard method for studying fish species' diversity and community ecology [32,44,45]. The IRI value for each species combines its frequency of occurrence (FO), percentage of total biomass ($P_W = W_{1-I}/W_T$), and percentage of total numbers of prey consumed ($P_N = N_{1-I}/W_T$):

$$IRI = (P_W + P_N)\, FO_{1-I} \times 10^4 \tag{1}$$

A species for which the value of IRI is >300 can be recognized as a dominant species; in this study, an IRI value of > 1000 was used to denote an absolute dominant fish species.

A Bray–Curtis similarity measure was also computed. This was composed of a similarity coefficient matrix based on the ratio of non-native fish to the total number of fish ($R_N = NN_{1-X}/N_T$), as well as the ratio of different non-native species in the total number of non-native fish species catch ($R_{NT} = N_{N1-X}/N_{NT}$). The $R_N$ and $R_{NT}$ were square-root-transformed in the analysis of the similarity matrix across the different rivers. After that, the similarity indices across the different rivers was used to perform a clustering analysis with Primer 5 software [46].

The numbers of non-native fish species (X) in the data from the Hainan Island rivers and the Continent coastal rivers was compared using one-way analysis of variance (One-Way ANOVA). The ratios of the most common non-native native fishes ($R_{NT} = NN_{1-X}/N_{NT}$) in the Hainan Island

rivers and the Continent coastal rivers were compared using a nonparametric test (two independent samples). The relationship between the temperature (mean minimum) and the abundance of dominant non-native fish were determined with linear regressions. All statistical analyses were performed with SPSS version 16.0, and the results were considered significant at $p \leq 0.05$.

## 3. Results

### 3.1. Species Composition

The total catch from the eight rivers comprised 98,887 individual fish, of total biomass 6525.65 kg. The total catch per river survey averaged 815.70 kg (range 542.62–1388.86 kg), and the average number of fish caught per river was 12,361 (range 6579–28,293) (Table 1). An average of 84 fish species were caught in each river ranging from 55 to 136.

A total of 11,832 individual non-native fish, of total weight 1531.90 kg, were caught in the 48 sites of eight rivers. Twenty non-native fish species were found: Nile tilapia, redbelly tilapia, suckermouth catfish, mrigal carp, North African catfish, marble goby *Oxyeleotris marmorata* (Bleeker), rohu *Labeo rohita* (Hamilton), streaked prochilod *Prochilodus lineatus* (Valenciennes), jaguar cichlid, Mozambique tilapia *Oreochromis mossambicus* (Peters), blue tilapia *Oreochromis aureus* (Steindachner), red pacu *Piaractus brachypomus* (Cuvier), sutchi catfish *Pangasianodon hypophthalmus* (Sauvage), redhead cichlid theraps maculicauda *Cichlasoma synspilum* (Regan), blackspot barb *Dawkinsia filamentosa* (Valenciennes), channel catfish *Ictalurus punctatus* (Rafinesque), redbreast tilapia *Coptodon rendalli* (Boulenger), galilaea tilapia *Sarotherodon galilaeus* (Linnaeus), mosquitofish *Gambusia affinis* (Baird and Girard), and walking catfish *Clarias batrachus* (Linnaeus).

The 20 non-native fish species collected represent 5 orders, 10 families, and 16 genera (Table 2); the orders most represented were the Perciformes, Siluriformes, and Cypriniformes, with 9, 5, and 4 species, respectively (Figure 2a). Of the non-native species, 16 are used for aquaculture, 3 in the ornamental fish trade, and 1 for biocontrol (Figure 2b). Seven of the non-native species originate from Africa, 5 from Southeast Asia, 4 from South America, 2 from Central America, and 1 each from South Asia and North America (Figure 2c). Of the 20 non-native species, 13 are omnivores and 7 are predators (Figure 2d).

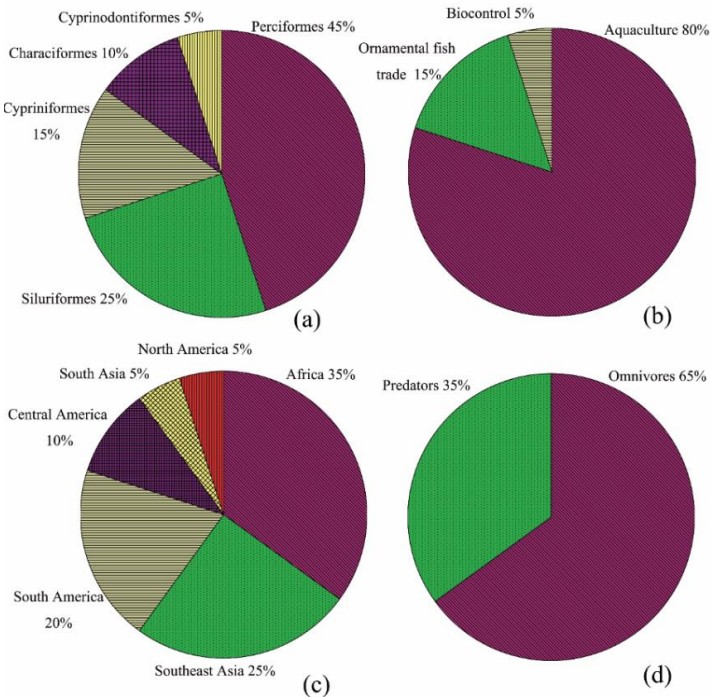

**Figure 2.** Percentages of non-native fish species found in the rivers of South China (**a**) by order, (**b**) by introduction vector, (**c**) by native region, and (**d**) by feeding strategies.

**Table 1.** Summary information of fish caught in surveys in eight rivers of South China: Nandujiang River (NDJ), Wanquanhe River (WQH), Changhuajiang River (CHJ), Jianjiang River (JJ), Moyangjiang River (MYJ), Dongjiang River (DJ), Xijiang River (XJ), and Beijiang River (BJ).

| River | River Length (km) | River Catchments (km$^2$) | Total Weight of Fish Collected (kg) | Total No. of Fish Collected | Weight Ratio of Non-Native Fish Species (%) | Ratio of Non-Native Fish Species (%) | No. of Non-Native Fish Species | No. of Sampling Sites | Mean Minimum Temperature (°C) |
|---|---|---|---|---|---|---|---|---|---|
| WQH | 160 | 3693 | 574.57 | 6220 | 37 | 16 | 14 | 10 | 17.23 |
| CHJ | 232 | 5070 | 542.62 | 28,293 | 46 | 7 | 11 | 9 | 16.55 |
| NDJ | 330 | 7177 | 1249.67 | 18,051 | 21 | 13 | 11 | 9 | 15.90 |
| MYJ | 199 | 6091 | 707.90 | 7315 | 37 | 22 | 8 | 5 | 12.81 |
| JJ | 230 | 9464 | 756.91 | 11,929 | 42 | 18 | 10 | 5 | 12.68 |
| XJ | 2100 | 353,120 | 1388.86 | 7900 | 5 | 2 | 8 | 3 | 11.05 |
| DJ | 500 | 35,340 | 641.62 | 6579 | 12 | 20 | 9 | 5 | 9.13 |
| BJ | 445 | 46,710 | 663.50 | 12,600 | 13 | 10 | 10 | 4 | 8.55 |

**Table 2.** List of non-native fish species found in eight rivers of South China: Nandujiang River (NDJ), Wanquanhe River (WQH), Changhuajiang River (CHJ), Jianjiang River (JJ), Moyangjiang River (MYJ), Dongjiang River (DJ), Xijiang River (XJ), and Beijiang River (BJ). "1" denotes presence of the species, and "0" its absence. "*" denotes the most dominant non-native fish in each river.

| Order | Family | Species | NDJ | WQH | CHJ | JJ | MYJ | DJ | XJ | BJ |
|---|---|---|---|---|---|---|---|---|---|---|
| Perciformes | Cichlidae | *Oreochromis mossambicus* | 1 | 0 | 1 | 1 | 0 | 0 | 0 | 0 |
| | | *Oreochromis niloticus* | 1 * | 1 * | 1 * | 1 * | 1 * | 1 | 1 | 1 |
| | | *Oreochromis aurea* | 1 | 1 | 0 | 1 | 1 | 1 | 0 | 1 |
| | | *Coptodon rendalli* | 0 | 1 | 0 | 0 | 0 | 0 | 0 | 0 |
| | | *Coptodon zillii* | 1 | 1 | 1 | 1 | 1 | 1 | 1 * | 1 * |
| | | *Sarotherodon galilaeus* | 0 | 1 | 0 | 1 | 0 | 1 | 0 | 0 |
| | | *Parachromis managuensis* | 1 | 1 | 1 | 0 | 0 | 1 | 0 | 0 |
| | | *Cichlasoma synspilum* | 1 | 1 | 0 | 1 | 0 | 0 | 0 | 0 |
| | Eleotridae | *Oxyeleotris marmorata* | 1 | 1 | 1 | 1 | 0 | 0 | 0 | 0 |
| Siluriformes | Loricariidae | *Hypostomus* sp. | 1 | 1 | 1 | 1 | 1 | 1 | 1 | 1 |
| | Clariidae | *Clarias gariepinus* | 1 | 1 | 1 | 1 | 1 | 1 | 1 | 1 |
| | | *Clarias batrachus* | 1 | 0 | 0 | 0 | 0 | 0 | 0 | 0 |
| | Ictaluridae | *Ictalurus punctatus* | 0 | 1 | 0 | 0 | 0 | 0 | 1 | 1 |
| | Pangasiidae | *Pangasianodon hypophthalmus* | 0 | 1 | 1 | 0 | 0 | 0 | 0 | 0 |
| Cypriniformes | Cyprinidae | *Cirrhina mrigala* | 0 | 0 | 0 | 1 | 1 | 1 * | 1 | 1 |
| | | *Labeo rohita* | 0 | 0 | 0 | 0 | 1 | 0 | 0 | 1 |
| | | *Dawkinsia filamentosa* | 0 | 1 | 0 | 0 | 0 | 0 | 0 | 0 |
| Characiformes | Prochilodontidae | *Prochilodus lineatus* | 0 | 0 | 0 | 0 | 1 | 0 | 1 | 1 |
| | Serrasalmidae | *Piaractus brachypomus* | 1 | 1 | 1 | 0 | 0 | 0 | 0 | 0 |
| Cyprinodontiformes | Poeciliidae | *Gambusia affinis* | 0 | 0 | 1 | 0 | 0 | 0 | 0 | 0 |

The non-native fishes collected in the greatest proportions were Nile tilapia (46.31%, *n* = 5,479), redbelly tilapia (28.52%, *n* = 3,375), jaguar cichlid (7.38%, *n* = 873), marble goby (6.61%, *n* = 782), and mrigal carp (4.09%, *n* = 484). Based on the values of IRI, Nile tilapia was the relative importance fish species in seven of the rivers (not in XJ), and could be recognized as dominant fish species in five of the rivers (NDJ, WQH, CHJ, JJ, and MYJ) (Figure 3). The redbelly tilapia was the relative importance fish species in four of the rivers (BJ, DJ, JJ, and MYJ), and could be recognized as dominant fish species in two of the rivers (BJ and DJ) (Figure. 3). Mrigal carp was the relative importance fish species in three of the rivers (BJ, DJ, and MYJ); marble goby was the relative importance fish species in CHJ and WQH; and jaguar cichlid was the relative importance fish species in NDJ and WQH (Figure 3).

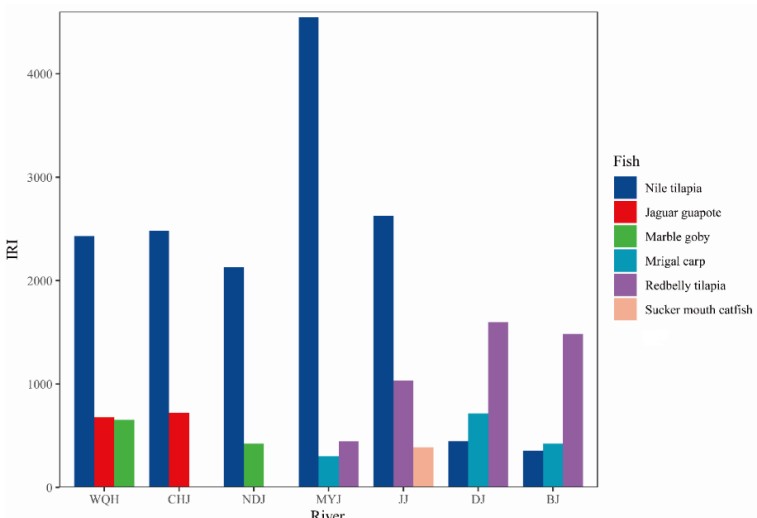

**Figure 3.** The index of relative importance (IRI) of dominant non-native fish species in the main rivers of South China. (Note: In XJ, no non-native fish species are recognized as a dominant species).

*3.2. Distribution Variations*

The weight ratio of non-native fish to all fish in the eight rivers averaged 26.6%, with a range of 4.7–45.5%, and the amount ratio averaged 13.6%, ranging from 2.1% to 22.5% (Table 1). The weight ratio of non-native fish to all fish in the 50 sites ranged from 0% to 79%, and the amount ratio averaged ranging from 0% to 71% (Figure 4). The number of non-native fish species in each river ranged from 8 to 14, with more non-native species found in the rivers of Hainan Island (i.e., NDJ, WQH, and CHJ), as compared with the Continent rivers (i.e., JJ, MYJ, DJ, XJ, and BJ) (*F* = 10.125, *p* = 0.019) (Table 1). The mrigal carp, rohu, and streaked prochilod were distributed only in the Continent rivers; the sutchi catfish, redhead cichlid, blackspot barb, mosquitofish, and walking catfish were distributed only in the Hainan Island rivers (Table 2).

According to the amount ratio of the different non-native species to the total number of fish collected ($R_N = NN_{1-X}/N_T$), the rivers could be grouped into the Continent river assemblage and Hainan Island river assemblage; the Continent river assemblage could be further divided into Continent coastal rivers (JJ and MYJ), the Pearl River system 1 (DJ and BJ), and the Pearl River system 2 (XJ) (Figure 5). In the Hainan Island rivers, Nile tilapia, marble goby, and jaguar cichlid were the dominant fish species, with Nile tilapia recognized as an absolute dominant and the most common non-native fish species (Figures 3 and 5). In the Continent coastal rivers, Nile tilapia, redbelly tilapia, suckermouth catfish, and North African catfish could be recognized as dominant species, with Nile tilapia being an absolute dominant and the most common non-native fish species (Figures 3 and 5). In Pearl River system 1, the dominant non-native fish species in the two rivers were the same, comprising the redbelly tilapia, mrigal carp, and Nile tilapia; redbelly tilapia was an absolute dominant and the most common non-native fish species (Figures 3 and 5). In Pearl River system 2, none of the non-native fish species could be recognized as dominant species (Figures 3 and 5).

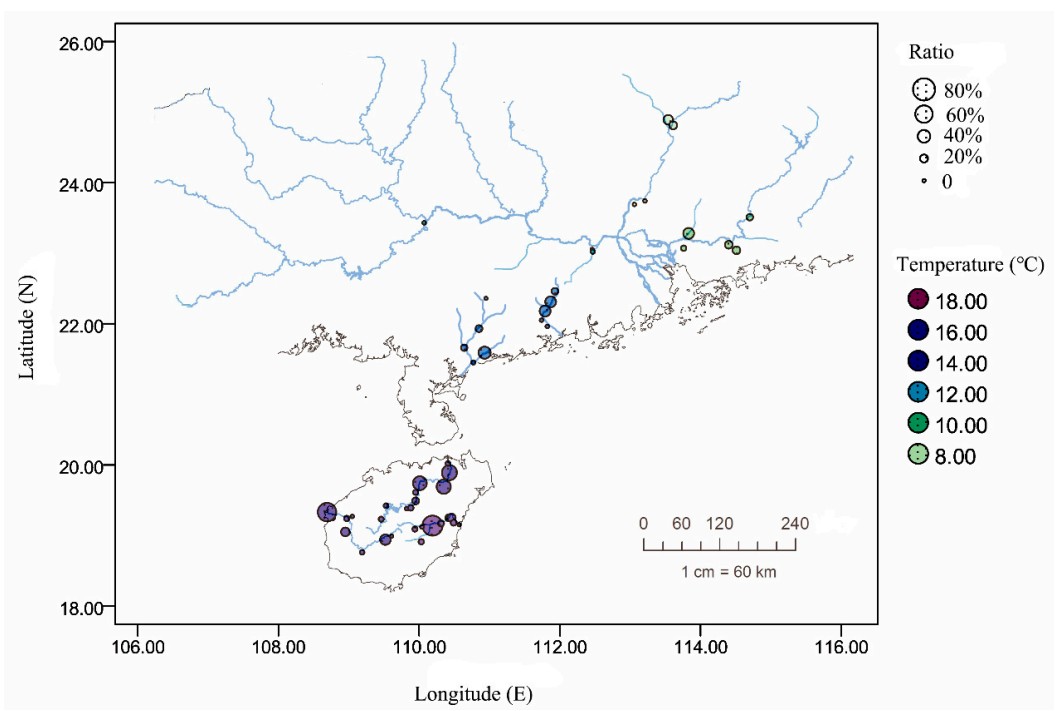

**Figure 4.** The distribution of non-native fish in 50 surveyed sites of south China (the bubble size per site signified the amount ratio of non-native fish species).

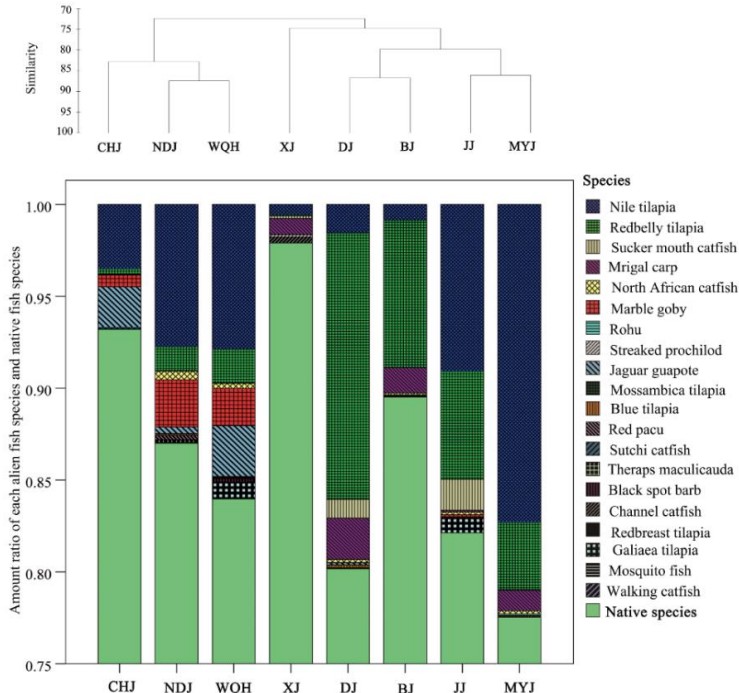

**Figure 5.** Ratios of the different non-native fish species to the total fish catch, used to classify the eight rivers surveyed.

According to the amount ratio of different non-native species to the total non-native fish species catch ($R_N = NN_{1-X}/N_{NT}$), the eight rivers could be divided into four assemblages: Hainan Island rivers, Continent coastal rivers (JJ and MYJ), Pearl River system 1 (DJ and BJ), and Pearl River system 2 (Figure 6). In the Hainan Island rivers, the Nile tilapia, marble goby, jaguar cichlid, and redbelly tilapia were the most common non-native fish (Figure 6). In the Continent coastal rivers (JJ and MYJ), Pearl

River system 1 (DJ and BJ), and Pearl River system 2, the redbelly tilapia, Nile tilapia, and mrigal carp were the most common non-native fish (Figure 6).

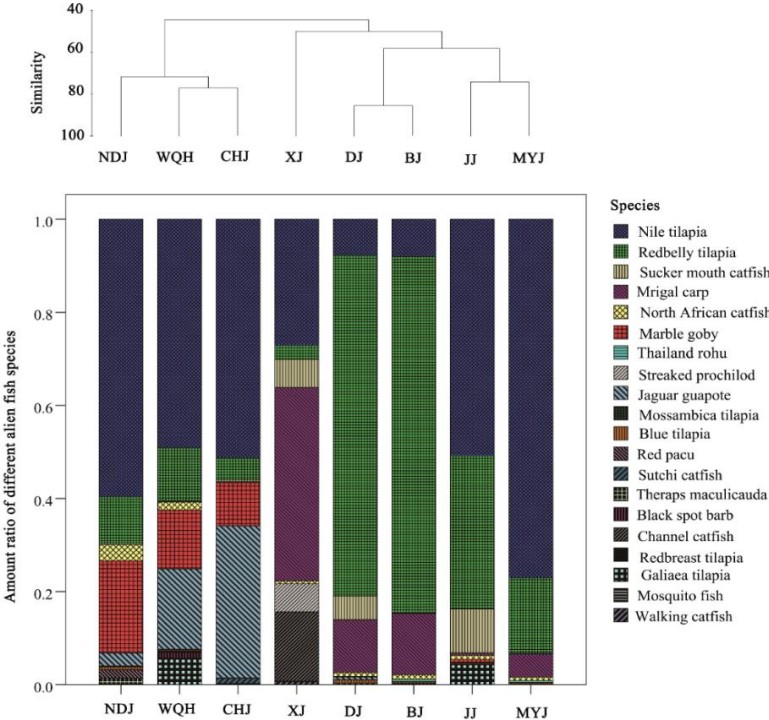

**Figure 6.** Ratio of each non-native species to the total non-native fish catch, used to classify the eight rivers surveyed.

Of the five most common non-native fish species, the amount ratio of marble goby, jaguar cichlid, and mrigal carp differed significantly between the Hainan Island rivers and Continent rivers, but the ratios of Nile tilapia and redbelly tilapia did not (Figure 7). Temperature was significantly correlated with the amount ratio of Nile tilapia and redbelly tilapia, with that of Nile tilapia increasing along the mean minimum temperature of the coldest month (Figure 8), and that of redbelly tilapia decreasing (Figure 8).

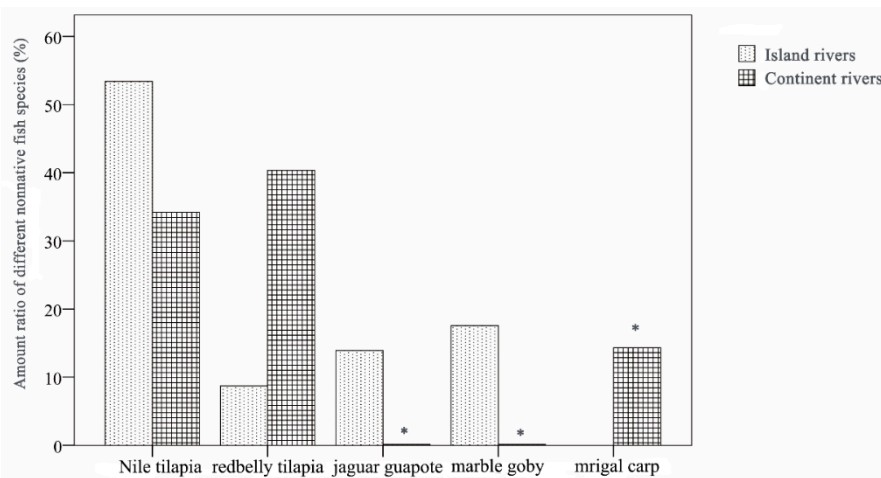

**Figure 7.** Ratios of the five-most-abundant non-native fish species in the Hainan Island rivers and the Continent rivers. An asterisk (*) indicates significant differences ($p < 0.05$) between the two assemblages.

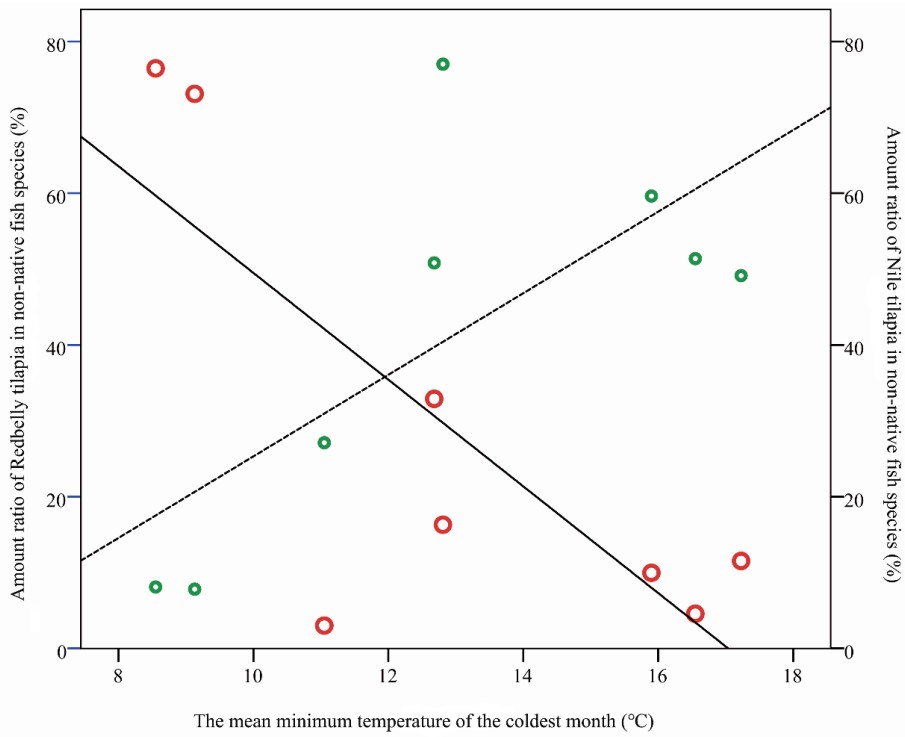

**Figure 8.** Relationship between the ratio of Redbelly tilapia (solid line, $p = 0.022$, $R^2 = 0.608$), Nile tilapia (dashed line, $p = 0.043$, $R^2 = 0.523$) and the mean minimum temperature in the coldest month (January 2018 data) across eight rivers in South China.

## 4. Discussion

### 4.1. Distributions of the Non-Native Fishes and Their Potential Impacts

Non-native fish species appear to be common and widespread in the freshwaters of South China, as they were recorded in all eight main rivers (Table 2). Although not all of the 20 species can be classified as invaders, all the rivers have been colonized by multiple non-native fish species.

Of the 20 non-native fish species found, the Nile tilapia, redbelly tilapia, mrigal carp, North African catfish, and suckermouth catfish are considered the most successful invaders because: (1) they were found in all of the studied rivers (Table 2); (2) they have established self-sustaining populations and can be considered as dominant fish (Figure 3); and (3) they have negatively impacted native fish species or water quality [9,47]. The marble goby and jaguar cichlid appeared to be successful invaders in the NDJ, WQH, and CHJ of Hainan Island, where they have established natural populations and impact native species through predation (Figures 3 and 7) [23,25,47]. These invasive fishes can cause a series of impacts on the region's biodiversity and economy. For example, the invasion of tilapia in Guangdong Province has caused economic losses in the cultured and capture fisheries, leading to species extinction, biodiversity loss, and deterioration of the natural aquatic environment [13].

### 4.2. Factors Influencing the Non-Native Fish Species Composition

Vectors of fish translocation and introductions include the aquarium trade, aquaculture rearing, fisheries stocks, and biological control [8,9,15,48]. Of the 20 non-native fish species found in South China, 16 are aquaculture species, and 6 of the 7 invasive fish species were introduced for aquaculture rearing (Figure 2b). Fish inadvertently introduced from aquaculture facilities typically have a strong ability to tolerate a range of environmental conditions, flexible habitat requirements, efficient reproductive strategies, and fast growth, as compared with fish species cultured for the ornamental trade [49,50]. For example, Nile tilapia is one of the most popular cultured fishes and it has become perhaps the

most widely distributed invasive species globally [13,51]. Therefore, vigilant management is needed to prevent the escape of cultured non-native fish species from aquaculture facilities.

All of the non-native fish species found in the eight rivers are native to tropical freshwaters, except for the channel catfish, which is native to North America, and all of the successful invasive fish species in these rivers of South China are native to the tropics [15,19] (Figure 2c). The rivers surveyed in this study have the warmest climates in China, and their environmental conditions are similar to the native habitats of the non-native fish species [42]. Furthermore, among the eight rivers, more non-native fish species were found on Hainan Island (NDJ, WQH, CHJ), where the rivers are tropical and warmer (Table 1). Although the region of the study contains most of the non-native fish species cultured in China, many species native to colder areas, such as largemouth bass *Micropterus salmoides*, were not found in the rivers. This result suggests that, in this region, alien fish species originating from regions with similar abiotic conditions will best survive and spread easily.

Thirteen of the non-native fish species found are omnivores, and 7 are predators (Figure 2d). Of the 7 successfully invasive species, the North African catfish, marble goby, and jaguar cichlid are effective predators, whereas the Nile tilapia, redbelly tilapia, mrigal carp, and suckermouth catfish are typical omnivores. Fish predators may succeed in a new environment when their prey are not adjusted to their particular style of predation [52]. Omnivores that forage on low-quality resources may succeed because such food resources are rarely limiting in aquatic systems [9,22,34,52]. Accordingly, a species' trophic position may be a key factor in its colonization and invasion capacity, and top predators and omnivores are most likely to be successful invaders in the rivers of South China.

### 4.3. Two Factors Most Affecting Distributions of the Non-Native Fish Species

The eight rivers surveyed could be divided to two big assemblages, the Continent rivers and Hainan Island rivers, according to the amount ratio, which reflects the geographical differences between continents and islands (Figure 3). The non-native fish species composition, numbers of species, and dominant species differed between the two assemblages of rivers (Tables 1 and 2; Figure 2). Of the five most-common non-native fish species found in the rivers, the amount ratios of 3 species significantly differed between the Hainan Island rivers and Continent rivers (Figure 7). The presence, survival, and spread of non-native fishes in specific natural waters usually results from intentional introductions, escapes of cultured fish, or disposals and ensuing self-spread [24]. This progression may be determined by local aquaculture conditions, market requirements, and policy support related to the geographical situation [7,30]. Hainan Island has become one of China's most important hatchery and culture areas for many non-native fish species owing to its relatively warm temperatures and similarities to the native habitats of the species reared [53]. As such, a greater number of non-native fish species was found on the island than was collected in the rivers on the continent, and several of these, namely the streaked prochilod, sutchi catfish, redhead cichlid, and blackspot barb, were found only on Hainan Island (Table 2). The island's isolation may block the spread of cultured non-native freshwater fishes after they escape or are discarded [39,51]. Thus, it seems that geographical isolation is an important factor distinguishing the species composition between the Hainan Island and the Continent rivers.

The ratio of non-native fish species to total fish species collected allowed further grouping of the Continent rivers into Continent coastal rivers (JJ and MYJ), the Pearl River system 1 (BJ and DJ), and the Pearl River system 2 (XJ) (Figure 3). Temperature also importantly affected the distribution patterns of the two widely distributed and common species—the Nile tilapia and redbelly tilapia (Figure 8). A warm climate makes the river basins of Hainan Island and the Continent coastal area important regions for Nile tilapia production [13,53–56]. Nile tilapia are indeed the most important cultured freshwater fish in these basins: 84.29% of freshwater fish production on Hainan Island constitutes tilapia farming, and tilapia are also the most important cultured fish in the basins surrounding JJ and MYJ, which produces ~ 18% of China's total tilapia production, while the Nile tilapia and its hybrids contribute to > 90% of the tilapia production [54,55]. The same climatic conditions in these river systems that promote the culture of Nile tilapia also support its establishment and spread once

it enters natural waters [21]. Furthermore, the Nile tilapia is a strong competitor in the warmer water because of its larger body size and fast growth rate [13,21]. Nile tilapia was the most common species in both the Continent coastal rivers and Hainan Island rivers, and could be recognized as an absolute dominant fish species in those river assemblages (Figures 3, 5 and 6). However, Nile tilapia is sensitive to temperature, with mortalities increasing at temperatures below 12 °C; thus, its abundance is less in colder rivers [21] (Figure 8). Although the second-most-abundant species, the redbelly tilapia is not popular in aquaculture owing to its small size and slow growth; however, it can tolerate lower temperatures than the Nile tilapia, and was thus more abundant in colder rivers and could be recognized as an absolute dominant fish species in the DJ and BJ (Figures 3 and 8). Previously, Gu et al. similarly found that the distribution patterns of these two most common invasive tilapia species were related to temperature conditions [21]. Likewise, Yu et al. noted similar circumstances for the distributions of the invasive mrigal carp and native mud carp *Cirrhinus molitorella* [25].

## 5. Conclusions

The non-native fish species are widespread and abundant in the rivers of south China, the establishment of them are clearly related to their introduction use, feeding strategy, and compatibility with their native habitat. Geographical isolation and temperature affected the non-native fish species composition, numbers of species, and dominant species in the region, and these two factors probably most influence the differences in their distributions.

**Author Contributions:** Conceptualization, D.E.G., J.W.W., and W.X.C.; methodology, Y.C.H. and J.W.W.; investigation, D.E.G., F.D.Y., M.X., X.D.M., D.L., Y.X.Y., H.W., Z.X.S., G.J.L., and Y.N.T.; data curation, D.E.G.; writing—original draft preparation, D.E.G.; writing—review and editing, D.E.G., J.W.W., and M.X.; project administration, Y.C.H.; funding acquisition, D.E.G. and J.W.W. All authors have read and agreed to the published version of the manuscript.

**Funding:** This research was funded by funding from the National Natural Science Foundation of China (Grant/Award Number 31500465), the National Key Research and Development Program of China (2018YFD0900705), the Pearl River Fisheries Research Institute – Chinese Academy of Fishery Sciences (CAFS) (ZC-2019-10), the Central Public-interest Scientific Institution Basal Research Fund – CAFS (2019HY-JC03; 2018SJ-ZH02; 2020TD17), and the Ministry of Agriculture and Rural Affairs (2130108).

**Acknowledgments:** We thank Yunjie Zhu, Xinwei Cai and the fishermen who helped with collecting fish.

**Conflicts of Interest:** The authors declare no conflict of interest.

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
