# Peer review of "The Species Composition and Distribution Patterns of Non-Native Fishes in the Main Rivers of South China"

_sustainability, doi:10.3390/su12114566_

Round 1
Reviewer 1 Report
This manuscript reports the results of an interesting field survey that catalogs and quantifies the extent of invasive fishes in Chinese river systems. This contribution is timely and interesting. It provides a very nice snapshot of the current situation as well as considers factors that might be contributing to the observations reported. In general it is also well written, although a final polish of the English would be nice.
I have relatively few concerns or suggestions on the manuscript. I outline these below in the order in which they appear in the manuscript.
Lines 86-89, 98-100: These sections appear out of place and disrupt the flow of the Introduction. I suggest that they be incorporated into the last paragraph of the Introduction.
Line 126: The biggest weakness of the paper is the description of the sampling protocol. In particular, it is not clear how many sites were sampled in each river, where they were located, and how sites were selected.
Line 156: Please explain how frequency of occurrence was calculated in this context and provide its abbreviation.
Lines 166-167: What was the unit of replication in the ANOVA?
Line 176: Since you are measuring in kilograms this is technically mass not weight.
Reviewer 2 Report
Overview;
It appears throughout the manuscript the authors have sought to make their sentences overly complicated, sometimes adding in jargon that makes the sentences circular and sometimes having sentences so broad that they don’t convey any specific message to the reader. The language for the paper needs refining to be concise and informative.
The methods are lacking some information on where results were obtained – i.e. feeding strategies, origins and routes of invasion.
Considering two of the three aims of the papers were around distribution I would have expected to see some maps of the non-native fish distributions displaying the 50 surveyed sites and potentially using bubble plots to signify either the biomass or abundance per site with bubble size. These could then have been coloured by the temperature of the river.
Abstract
20 – 21 – ‘the same climatic conditions that facilitate the culture of some non-native fishes may also support their invasive potential’ this statement is so vague it does not actually make a point. Anywhere in the world cultured fish ‘may’ be in the temperature that enables them to become invasive. A more informative statement would be ‘Tilapia, catfish and carp are regularly cultured in South China which is similar to their climate in xxx area’.
21 – 22 A more concise wording would be ‘We systematically collected fish from eight main rivers…’
22 ‘analysis’ should be ‘analyse’
23 – 25 The two halves of this sentence don’t really align, the first part says ‘widespread’ therefore is about spatial distribution and the second part is about abundance/ proportions of fish, either remove the semi colon and instead insert a comment about the ‘abundance’ or include a comment in the second part about the range from a – b and c – d with maybe spatial reference with latitude and longitude.
26 insert ‘the’ between ‘while’ and ‘other’
32 replace comma with fullstop and capitalise ‘their’
Introduction
39 remove ‘and’ before economic as this is in the middle of a list, only the final part of a list should begin with and
40 the mention of parasites here is vague, do you mean they bring new parasites? Help to alleviate parasite load in existing native populations? None of the references [1, 2] relate to parasites, need a reference for each of the examples mentioned. E.g. https://link.springer.com/content/pdf/10.1007/s10530-020-02253-1.pdf
53 – 56 If the phenomenon is relevant to both invasive and non-invasive non-native fish then this second sentence can be removed and the first sentence delete ‘invasive’
74 delete ‘introduction’ as it is repeated here and line 71, it doesn’t need mentioning twice, it’s obvious that an introduction is necessary for establishment. Simplify sentence to; ‘For effective establishment the non-native fish must survive and be dispersed by either natural or anthropogenic means.’
75 – 76 ‘anthropogenic spread’ by it’s very definition is from ‘human activity’ so this can be simplified by deleting ‘anthropogenic’
78 delete ‘differences’ and replace with ‘differs’
78 insert ‘can be’ between ‘which’ and ‘illustrated’
78 – 79 the whole sentence can be deleted ‘The vector of introduction is significant in the ingress of non-native fish species into the rivers of South China’ this doesn’t give any information, vectors are mentioned in the previous sentence and this comes with no citation to follow up this information.
86 The citation 31 – Marchetti et al. 2004 does not mention trophic niches, this could be a useful citation relating to the trophic niches of invaders https://besjournals.onlinelibrary.wiley.com/doi/full/10.1111/1365-2656.12996
88 see comment on line 206 about inappropriate use of the term ‘trophic position’ for the data presented in your results.
90 – 91 It is not clear what the point of this sentence is. Reword. The citation is about faunal homogenisation which does not seem related to the sentence.
92 consider removing ‘biogeographical conditions’ if biogeography is the distribution of animals in space and time then by it’s very definition the composition of fish species will vary by biogeography, I don’t think this conveys any tangible meaning to the reader and ‘environmental conditions’ are mentioned on the next line. If this is kept, this needs a rebuttal.
97 ‘growing’ replace with ‘growth’
98 it is not clear that any of those listed references support the reproductive strategies of fish determining introduction, culture and trade. Instead consider; https://onlinelibrary.wiley.com/doi/full/10.1111/j.1095-8649.2003.00210.x?casa_token=e3EzRnabRysAAAAA%3AknkFeitEUoyXUbHNasT6c6hd3w3Esjx1mwGqeubI7QrsAD3tNsmqueOwwy49EV8swrKzVbkPw1F_2w
104 remove ‘main rivers of South China’ already mentioned in previous sentence.
105 Not clear how you can investigate ‘fish species’ by ‘fish fauna’ and nothing in the methods or results mentions ‘fish fauna’ suggest deletion.
Methods
110 – 111 Simplify sentence to ‘Fish were surveyed from 2016 to 2018 at 50 sites on rivers in three provinces…’
114 – 116 instead of listing values, add these to Table one and just cite here that ‘The river lengths vary from 160 to 2,100 km and the catchments from 3,693 to 353,120 km2 (Table 1).
Seems unclear as to why the smaller catchments have more sampling points that the larger catchments i.e. CHJ has 7 by looking at Fig 1 but XJ only has 3, and whey are there two sampling points outside of the coloured regions? More information needed on the distribution of the 50 sampling sites and why they are chosen.
126 Dates needed – what is the first and second half of year January – June? Consider supplementary information with specific dates for each site and which region they belong to, in a table.
131 what size gillnets?
132 how were the fish weighted and to what accuracy?
165 Which version of Primer?
167 this is the first mention of the ‘coastal rivers’, why are these being compared with ‘Hainan rivers’? This needs explaining in the introduction or methods.
170 – 172 refocus this sentence to the point of interest which is the comparison, not the method. i.e. ‘The relationship between the temperature (mean minimum) and the abundance of dominant non-native fish were determined with linear regressions.’ ‘in the coldest month (data for January 2018)’ is not needed as explained in lines 147 – 151
Results
178 – 179 reword for conciseness ‘An average of 84 fish species were caught in each river ranging from 55 to 136.’
Table 1. It would be helpful if these were in the order of latitude. See suggestion from 114 for columns to add to this table. Is it necessary to report two decimal places for the percentages? I would think whole numbers is sufficient for this level of accuracy. This would also be a good place to display the temperature data per river.
200 a common way to present presence absence is with 1 and 0, also the IRI takes up a large portion of methods but only features once in the results, the most dominant non native fish could easily be marked in this table for each river. This table would be much more informative if it showed the percentage of fish made up of each species instead of just presence absence.
Nothing in the methods explains where the information on feeding strategy comes from. This needs to be included. Why are the only feeding methods shown as predators and omnivores and not herbivore or detritivore – e.g. Galileae tilapia only eats detritus according to Fishbase.se therefore why would that be classed as an omnivore? Feeding method could be listed in Table 2 instead of summarising only across the whole of South China.
206 Figure 2 – ‘trophic position’ is not an accurate description of what is presented – consider changing to ‘feeding strategy’ or ‘trophic level’, trophic position is commonly used in studies that quantify trophic position with isotopes. Nothing in methods explains where the information on introduction vectors came from, are these all well documented? Surely these are just estimated introduction vectors or has their introduction been documented – more information needed. Same comment applies re lack of methods on origin of fish.
209 – 215 the use of the term ‘the dominant fish’ to refer to multiple different species within one river does not make sense, the most dominant fish is the most numerical or of greatest weight. Use the terms relative importance instead of dominance. And use ‘the dominant’ only to refer to the ‘absolute’ dominant.
217 – 218 Figure 3. remove repetition of the y-axis label by just featuring on the left hand graph of each row. Y axis values are not legible and don’t appear to be the same, either make them comparable or mention in the legend that the y-axes are variable. As the map is largely obstructed from view and a map has already been presented, why not just show these 7 graphs in a grid. Also the textured fill on the plots is not easily distinguished – use simpler fills i.e. all black, black lines, black dots, black criss cross etc. Or just use block colours if colours are wanted. Legend should mention IRI.
218 ‘could’ or ‘are’?
220 insert ‘to native fish’ after ‘fish’
221 only cite (Table 1) once and at the end of the sentence.
Figures 7 and 8 could be displayed on the same figure to save space. Just give one a solid line and one a dashed line and put the R squared and P values in the legend instead of in the graph.
Discussion
278 Not sure how you can be citing all references 6 to 43, assume this is an error. Where is the evidence for the water quality impacted? Clearer if you put the reference direcly after the fact that you include. Also paper 43 is only about Nile tilapia and this sentence suggests those 5 species all have these impacts. Reword this sentence for clarity.
283 – exaction? Is extinction meant here or extirpation?
305 I recommend using the word ‘abiotic’ instead of ‘geographical’
345 ‘fast growth rate’ instead of ‘faster grow rate’
Conclusions
357 – 359 This sentence doesn’t make sense, it needs to be reworded
359 I do not see how the authors can conclude that ‘particularly their ecological trophic position’ is important in determining non-native fish species when all that was presented was a single pie chart showing ‘predators’ and ‘omnivores’. Also see comment from line 206 about inappropriate use of terminology.
